# Haemodynamic Parameters Underlying the Relationship between Sarcopenia and Blood Pressure Recovery on Standing

**DOI:** 10.3390/jcm13010018

**Published:** 2023-12-19

**Authors:** Eoin Duggan, Silvin P. Knight, Feng Xue, Roman Romero-Ortuno

**Affiliations:** 1Discipline of Medical Gerontology, School of Medicine, Trinity College Dublin, D02 R590 Dublin, Ireland; 2Falls and Syncope Unit (FASU), Mercer’s Institute for Successful Ageing, St James’s Hospital, D08 KC95 Dublin, Ireland

**Keywords:** sarcopenia, delayed blood pressure recovery, orthostatic hypotension, orthostatic haemodynamics, orthostasis

## Abstract

Background: Sarcopenia, delayed blood pressure (BP) recovery following standing, and orthostatic hypotension (OH) pose significant clinical challenges associated with ageing. While prior studies have established a link between sarcopenia and impaired BP recovery and OH, the underlying haemodynamic mechanisms remain unclear. Methods: We enrolled 107 participants aged 50 and above from a falls and syncope clinic, conducting an active stand test with continuous non-invasive haemodynamic measurements. Hand grip strength and five-chair stand time were evaluated, and muscle mass was estimated using bioelectrical impedance analysis. Participants were categorised as non-sarcopenic or sarcopenic. Employing mixed-effects linear regression, we modelled the effect of sarcopenia on mean arterial pressure and heart rate after standing, as well as Modelflow^®^-derived parameters such as cardiac output, total peripheral resistance, and stroke volume, while adjusting for potential confounders. Results: Sarcopenia was associated with diminished recovery of mean arterial pressure during the 10–20 s period post-standing (β −0.67, *p* < 0.001). It also resulted in a reduced ascent to peak (0–10 s) and recovery from peak (10–20 s) of cardiac output (β −0.05, *p* < 0.001; β 0.06, *p* < 0.001). Furthermore, sarcopenia was associated with attenuated recovery (10–20 s) of total peripheral resistance from nadir (β −0.02, *p* < 0.001) and diminished recovery from peak (10–20 s) of stroke volume (β 0.54, *p* < 0.001). Notably, heart rate did not exhibit a significant association with sarcopenia status at any time interval post-standing. Conclusion: The compromised BP recovery observed in sarcopenia appears to be driven by an initial reduction in the peak of cardiac output, followed by attenuated recovery of cardiac output from its peak and total peripheral resistance from its nadir. This cardiac output finding seems to be influenced by stroke volume rather than heart rate. Possible mechanisms for these findings include cardio-sarcopenia, the impact of sarcopenia on the autonomic nervous system, and/or the skeletal muscle pump.

## 1. Introduction

Humans stand on average 60 times per day [1]. One of the key challenges associated with orthostasis is the maintenance of blood pressure (BP) in the upright position. Gravity leads to a pooling of blood in the lower limbs and below the level of the heart and therefore to a reduction in intrathoracic blood volume [2]. As a consequence, this leads to a decrease in central venous pressure and therefore stroke volume (SV), resulting in a decline in cardiac output (CO) [3]. Without counter-regulatory measures, a precipitous fall in mean arterial pressure (MAP) would occur [4]. 

A number of mechanisms are involved in the recovery of BP on standing. Carotid and cardiopulmonary baroreceptors detect a fall in BP triggering a reduction in cardiac vagal activity and an increase in sympathetic activity leading to an increase in heart rate (HR), which limits the fall in CO (caused by the drop in SV) [5]. Total peripheral resistance (TPR) increases via sympathetic mediated vasoconstriction in splanchnic and peripheral blood vessels [6]. Finally, the rhythmic activity of the skeletal muscle pump of the lower limbs may also improve venous return and subsequent SV and CO, thus helping maintain MAP [7].

Impairment of these mechanisms with ageing or disease may result in a delayed recovery of BP or a failure of BP to recover—orthostatic hypotension (OH) [8]. In general, a ‘classic’ non-recovery OH pattern observed at three minutes post-standing, especially when accompanied by a reduced or absent HR increase, is more likely to be ‘neurogenic’ or largely caused by either primary or secondary autonomic failure [9]. However, the importance of causes such as heart failure, medications, and hypovolaemia are increasingly recognised in non-neurogenic presentations [10].

A potential contributor to OH is sarcopenia, a clinical condition of low muscle mass and function [11] that is prevalent in older people [12] and has been associated with many adverse outcomes [13]. Previous research has demonstrated an association between classic OH, assessed with intermittent BP measurement, and sarcopenia [14,15]. We previously found an association between beat-to-beat orthostatic BP recovery and both probable sarcopenia in a population study [16] and sarcopenia in a clinical setting [17].

However, the potential mechanisms underlying these associations have not been investigated and remain unclear. Departing from the well-established physiology equations: MAP=CO×TPR and CO=SV×HR [18], the aim of this study was to explore how the relationship between orthostatic BP recovery and sarcopenia may be mediated in haemodynamic terms.

## 2. Materials and Methods

Participants were recruited from the Falls and Syncope Unit at Mercer’s Institute for Successful Ageing in St James’s Hospital, Dublin, Ireland [19]. Typically, participants had been referred to the unit by their general practitioner due to a recent history of falls, syncope, pre-syncope, light-headedness, or dizziness. Criteria for inclusion in this study were as follows: age 50 years or older, able to provide written informed consent, able to mobilise independently (with or without aid), and able to transfer independently or with minimal assistance of one person from lying to standing. Exclusion criteria were as follows: history of confirmed or presumed neurogenic OH or contraindication to bioelectrical impedance analysis (BIA) (e.g., presence of an indwelling electronic device such as a cardiac pacemaker). Ethical approval for this study was granted from the Tallaght University Hospital/St. James’s Hospital Joint Research Ethics Committee (Project ID: 0221; approval date: 4 May 2021), and approval was also granted by St James’s Hospital Research & Innovation Office (Reference: 6567, approval date: 26 July 2021). All participants in this study provided explicit, written informed consent. This study adhered to the World Medical Association Declaration of Helsinki on ethical principles for medical research involving human subjects.

Participants’ medical history and use of medications were obtained from sources including self-report, electronic health record review, and general practitioner correspondence. Cardiovascular medication use was defined as regular use of one or more of the following medications (coded using the Anatomical Therapeutic Chemical Classification (ATC)): anti-arrhythmics (ATC C01), anti-hypertensives (ATC C02), diuretics (ATC C03), vasodilators (ATC C04), beta-blocking agents (ATC C07), calcium channel blockers (ATC C08), or agents acting on the renin–angiotensin system (ATC C09). Psychotropic medication use was defined as taking one or more of the following: anti-epileptics (ATC N03A), anti-psychotics, anxiolytics, hypnotics or sedatives (ATC N05), or anti-depressants (ATC N06A).

Hand grip strength (HGS) was assessed using a Jamar hydraulic hand dynamometer (Performance Health, Cedarburg, WI, USA). The maximum value of two seated, consecutive measurements taken on the left and right hands was used. Values were rounded to the nearest 2 kg as per the precision of the device. The five-chair stand test time (5CST) was measured as the time to the nearest centi-second taken for a participant to stand up and sit back down five times from a standard chair (approximate height 43 cm), as fast as possible. Height was measured to the nearest 0.01 m with a Seca 222 Stadiometer (Seca Ltd., Birmingham, UK). BIA was performed with a TANITA^®^ DC-430 MAP Body Composition Analyser (Tanita Europe, Amsterdam, The Netherlands). Participants stood barefoot on the scale, which also provided weight measurement to the nearest 0.01 kg. Participants were asked to remove their outerwear and empty their pockets, and 0.5 kg was entered as a standard tare value for clothing. Appendicular skeletal muscle mass was estimated using the Sergi equation [20]. The European Working Group on Sarcopenia in Older People (EWGSOP) revised criteria for sarcopenia (HGS of less than 27 kg in men and 16 kg in women and/or 5CST greater than 15 s, with muscle mass less than 20 kg in men and 15 kg in women) were used [21].

Participants underwent an active stand test with continuous non-invasive beat-to-beat BP measurement with Finapres^®^ Nova (Finapres Medical Systems, Amsterdam, The Netherlands). The active stand test consisted of a supine signal calibration and resting phase of 5–10 min, followed by standing as quickly as possible and remaining standing quietly for 3 min [22]. Signal calibration was performed using oscillometric brachial calibration and the Physiocal function on Finapres^®^ Nova with the height correction unit adjusting for differences in hydrostatic pressure. Finapres^®^ Nova automatically measures HR and calculates MAP per beat as the integral of the arterial pressure waveform.

Beat-to-beat signals for the haemodynamic parameters CO, TPR, and SV are derived by Finapres^®^ Nova using the built-in Modelflow^®^ algorithm. The Modelflow^®^ algorithm calculates the aortic flow waveform by simulating a non-linear three-element model of aortic input impedance [23]. This is integrated to compute SV, and from this, CO is calculated by multiplying by HR [24]. TPR is then derived from MAP divided by CO.

Values were exported from Finapres^®^ Nova software and analysed using custom-written software in MATLAB^®^ version 9.14 (The MathWorks, Inc., Natick, MA, USA) in accordance with previously published recommendations [25]. The baseline signal was taken as the mean of 60 to 30 s before standing. For the period from 0 to 180 s after standing, a ±5 s moving average filter was applied to the signals. The relative change in haemodynamic parameters, at 10 s time intervals from 0 to 180 s, was then calculated by subtracting the baseline from the absolute values.

Statistical analysis was performed with Stata^®^ version 15.1 (StataCorp LLC, College Station, TX, USA). To assess the relationship between sarcopenia and each of the haemodynamic parameters, we used multi-level mixed-effects models, as previously employed by our group and others [16,17,26,27]. The fixed and random effects accounted for the repeated measures within participants. Piecewise linear splines were used to model five time intervals: 0–10, 10–20, 20–30, 30–40, and 40–180 s after standing. Residual variance was modelled with a first-order autoregressive process to account for the strong correlation between adjacent timepoints. The linear splines were entered into the model as independent parameters along with the interaction of sarcopenia status (no sarcopenia = 0; sarcopenia = 1) and potential confounders as covariates. The potential confounders considered were age, sex, diabetes, hypertension, cardiovascular medications, and psychotropic medications.

## 3. Results

Of the 123 participants recruited to this study, 5 had a contraindication to or declined BIA, 7 had probable or confirmed neurogenic OH, 1 had no active stand test, and in 3, the beat-to-beat haemodynamic signals were of poor quality. This left a final analytical sample size of 107 with a mean age of 69.7 years (standard deviation of 10.4 years), with 61 (57%) being women. Further characteristics of this sample, grouped by sarcopenia status, are presented in Table 1.

The beta coefficients with confidence intervals and *p*-values, at each of the five time segments, from the adjusted mixed-effects models for the relationship between sarcopenia status and each of the haemodynamic parameters are shown in Table 2. Plots of the marginal effects of the means for the two groups are shown in Figure 1 for each of the haemodynamic parameters.

The recovery in MAP in the 10–20 s post-stand period was significantly attenuated in the sarcopenia group compared with the no sarcopenia group as evidenced by the highly statistically significant negative beta coefficient, at a time when the overall MAP trend was upwards. 

CO for the sarcopenia group had an attenuated initial rise to peak in the 0–10 s period, shown by the significant negative coefficients when overall CO was rising. This was followed by an attenuated recovery of CO from peak, for the sarcopenia group, in the 10–20 s period, demonstrated by the significant positive beta coefficient at a time when CO was returning to baseline.

TPR showed a significantly attenuated recovery in the 10–20 s period after standing for the sarcopenia group compared with no sarcopenia, again as indicated by a significant negative coefficient at a time when overall TPR was recovering.

For SV, the sarcopenia group had an attenuated recovery in the 10–20 s period, demonstrated by a significant positive coefficient. For HR, no significant differences were seen between the two groups at any of the time intervals.

## 4. Discussion

In this study, we explored the relationship between sarcopenia status (present vs. absent) and the haemodynamic response to standing. We found that sarcopenia was associated with an attenuated recovery of MAP during the 10–20 s period after standing, in keeping with our previous findings regarding systolic and diastolic BP [17]. We expanded upon our previous findings by examining the components of MAP, namely CO and TPR, demonstrating early post-standing changes in both factors that may drive the alterations observed in MAP. Furthermore, CO is determined by SV and HR, and we observed significant changes in SV in the 10–20 s period after standing but no significant difference in HR between sarcopenia and non-sarcopenia groups. These results are important as they provide insights into the possible underlying pathophysiology of delayed BP recovery in older adults with sarcopenia, and this may have implications for the clinical management of this condition. Our findings are strengthened by a well-characterised participant cohort, enabling adjustments for potential confounders. Additionally, the use of continuous beat-to-beat haemodynamic measurements allows for a non-invasive and precise characterisation of the haemodynamic response to orthostasis.

The finding of an attenuated rate of recovery of MAP in the 10 to 20 s period after standing is in keeping with our previous study, which also found an attenuated rate of systolic and diastolic BP recovery in both probable and confirmed cases of sarcopenia during the same timeframe [17]. To the best of our knowledge, apart from our previous study, no previous studies had examined the relationship between sarcopenia and the early BP response to standing. However, one study [28] found an association between 5CST (a component of the EWGSOP criteria for probable sarcopenia) and diastolic BP recovery, 30 to 60 s after standing, with a longer 5CST being associated with worse diastolic BP recovery. Another study [29] found that those in the slowest category for 5CST had significantly higher odds of initial orthostatic hypotension in the first 15 s after standing. Together with our study, they add weight to the hypothesis that muscle function and mass may be linked to BP recovery early after standing.

CO showed a significantly attenuated initial increase in the sarcopenic group when compared with the non-sarcopenic. The sarcopenic group also showed an attenuated recovery from peak CO in the 10 to 20 s period. Thus, we can surmise that the changes in MAP were related to the attenuated CO response in the initial 10 s contributing to subsequent slower MAP recovery in the 10 to 20 s period. The initial increase in CO on standing followed by prompt recovery seen in the non-sarcopenic group has been previously established in healthy participants [30,31]. While no previous data had been reported in subjects with sarcopenia, Wieling et al. [32] found that older participants had a less pronounced rise in CO to peak after standing when compared with younger participants, similar to our finding in CO in the same period. The effects of ageing on skeletal muscle may have played a role in these findings. On the other hand, Van Wijnen and colleagues [33] found that in a group with delayed BP recovery, there was no significant difference on average in CO compared with a normal BP recovery group, but there was a large interparticipant variation in the delayed BP recovery group; yet, muscle mass or strength was not assessed in this study, and since the participants were 18 years and older, the likelihood of sarcopenia within this cohort is low.

Examining the TPR response, it is known that in healthy adults, TPR falls initially due to a number of mechanisms related to the muscular effort of standing, before recovering from nadir [34]. The attenuated recovery of TPR in the sarcopenia group in the 10–20 s period is similar to that of MAP, suggesting that the attenuated MAP recovery is driven at least partially by TPR [6]. Indeed, Van Wijnen et al. [33] found that TPR recovery was impaired in those with delayed BP recovery compared with normal BP recovery, while changes in CO were not different, suggesting that TPR was the primary driver of delayed BP recovery in that group. Decreased TPR leading to OH has been previously reported in diabetics [35], and Pérez-Denia et al. [36] found impaired TPR recovery in those with multimorbidity and suggested that this might drive impairments in BP recovery. Again, in these two studies, muscle parameters were not examined, but we know that both multimorbidity [37] and diabetes [38] are related to sarcopenia, and so it may be one of the factors involved.

When breaking down CO into its constituents, SV and HR, we observed that SV showed an attenuated recovery in the 10–20 s period, while there were no significant differences in HR. This suggests that CO changes were driven by SV rather than HR. This finding is somewhat unexpected, as one might anticipate an increase in HR in individuals with sarcopenia to compensate for the potential reduction in venous return due to impaired skeletal muscle pump function, which can result in decreased SV. The association between sarcopenia and SV/HR after standing has not been previously investigated, but the response of SV to standing has been previously shown to differ between younger and older subjects. Wieling et al. [32] demonstrated stable SV with a large increase in HR in the initial 7 s after standing in the younger group, while in the older group, there was less of an HR increase with the initial fall in SV followed by transient recovery and subsequent further fall. As before, although muscle factors were not included, it is possible that they played a role in the differential response seen in older people. 

There are several potential mechanisms underlying our findings. Specifically, in this sample, the effects on CO seemed primarily driven by SV rather than HR. It is established that SV is determined by preload, contractility, and afterload [39]. Afterload is largely determined by MAP itself, thereby acting as a negative feedback mechanism [40]. Contractility is determined in part through preload via the Frank–Starling mechanism, as well as intrinsic factors [41]. It is here that the influences of sarcopenia on the heart muscle might act. Cardio-sarcopenia is a relatively novel concept [42], especially as traditionally hypertrophy rather than wasting of the heart is seen with ageing. However, a small number of studies have shown reductions in the left ventricular mass in sarcopenia [43,44]. If correct, one might hypothesise that cardio-sarcopenia can lead to decreased contractility, leading to decreased SV, in turn impacting CO and subsequently leading to delayed recovery of MAP as seen in our study. When examining preload, it is influenced by the end-diastolic volume, which is influenced by venous return [40]. It is in this aspect that the skeletal muscle pump may play a role [7], potentially contributing to the effects of sarcopenia. This impairment in preload might subsequently affect SV, consequently leading to impaired MAP recovery as described in the pathway above. With regard TPR, this may be an effect of sarcopenia on the autonomic nervous system. It has been shown that general muscle sympathetic nerve activity (MSNA) is proportional to the muscle mass involved [45]. Therefore, those with sarcopenia might have reduced MSNA leading to impaired recovery of TPR and resultant delayed recovery of MAP.

Of course, there may be other mechanisms at play. Other syndromes such as frailty may be involved, although the underlying pathophysiology between sarcopenia and physical frailty is increasingly recognised as likely being common [46]. Impairment of the exercise pressor reflex [47] may be involved, although this is less likely as no significant differences in HR were found in our study. In a cross-sectional analysis, we cannot assert causality. However, it is more plausible that the pathway from sarcopenia to impaired BP recovery is the causal direction rather than the reverse pathway from impaired BP recovery to sarcopenia. There is the possibility of a positive feedback loop in that those with impaired BP recovery leading to falls may restrict activity due to fear of falling, which could lead to sarcopenia. We were unable to adjust for this pathway in our analysis and so acknowledge it as a limitation. 

The relatively small sample size in this study is another limitation. We conducted a post hoc power analysis for the difference in predicted mean change in CO at 10 s for the non-sarcopenic group vs. the sarcopenic and found that the power was 0.72 for an alpha of 0.05 (means 1.19 L/min and 0.57 L/min and standard deviations 0.85 L/min and 0.87 L/min, respectively). It appears, however, that the repeated-measures nature of the mixed-effects models is sufficiently sensitive to change to allow us to discern subtle differences in slopes of recovery of the haemodynamic parameters after standing. 

Other limitations of this study include the inherent limitations of BIA in the estimation of muscle mass [48] and the lack of information on timing of medications with respect to the active stand test. Furthermore, the use of Modelflow^®^ derived haemodynamics has inherent limitations [49]. While the gold standard would be invasive measurement [50], conducting this procedures would be neither practical nor ethical in this population. Doppler echocardiography can be used; however, Modelflow^®^ has been shown to be an acceptable alternative [51] especially when relative haemodynamic changes are required rather than absolute [52], as in our design. Finally, as participants were recruited from patients referred to a falls and syncope unit, they may not be representative of the general population, thus limiting the generalizability of our results.

Our results are important for a number of reasons. Firstly, from a pure pathophysiological and scientific point of view, they are novel. To the best of our knowledge, no previous study has utilised our methodology to address the question as to how sarcopenia can contribute to delayed BP recovery in older individuals. Secondly, results are of clinical importance. Through the use of beat-to-beat BP monitoring, impaired BP recovery has become increasingly recognised [53,54] and has been shown to independently predict falls, fractures, and mortality [55,56,57]. Our results link sarcopenia with impaired BP recovery, and this, in turn, can potentially be implicated in subsequent falls and fractures. Importantly, sarcopenia is potentially reversible, and multicomponent interventions that improve sarcopenia [58] could also be used to ameliorate delayed BP recovery. This, of course, underlines the importance of identifying sarcopenia in the falls and syncope unit in the first place [59]. Pressor agents such as fludrocortisone and midodrine are routinely prescribed for OH and delayed BP recovery [60] despite their off-label use and potential side effects [61]. Perhaps increasing use of multicomponent intervention as well as training in physical counterpressure manoeuvres [62] would lessen the use of these medications and also benefit cases of combined supine hypertension and OH [63], where physical interventions can help both.

## 5. Conclusions

We observed sarcopenia to be associated with an attenuated rate of recovery of MAP early after standing. This appeared to be driven by initial blunted peak of CO followed by attenuated recovery of CO from peak and TPR from nadir. The CO finding appeared to be driven by SV rather than HR. There are a number of potential mechanisms implicated, including cardio-sarcopenia, the effects of sarcopenia on the autonomic nervous system, and the skeletal muscle pump, and further research is needed to clarify the mechanisms involved and their relative contributions.

## Figures and Tables

**Figure 1 jcm-13-00018-f001:**
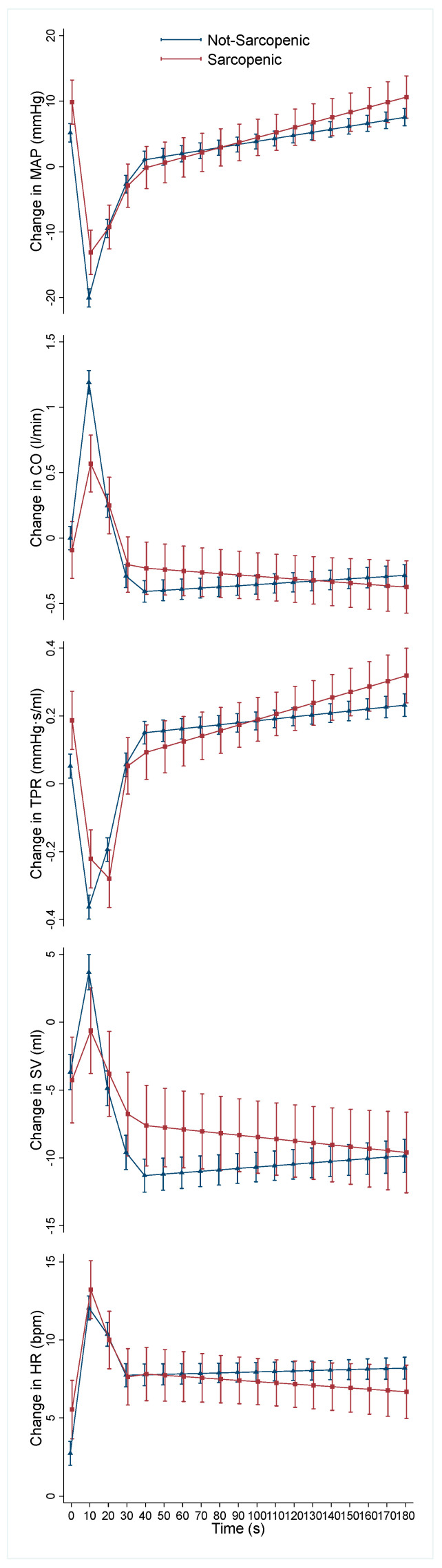
Predicted means and standard error of the mean from mixed-effects models for change in mean arterial pressure (MAP; mmHg), cardiac output (CO; L/min), total peripheral resistance (TPR; mmHg·s/mL), stroke volume (SV; mL), and heart rate (HR; beats per minute) from baseline after standing grouped by sarcopenia status. Models were adjusted for age, sex, diabetes, hypertension, cardiovascular, and psychotropic medications.

**Table 1 jcm-13-00018-t001:** Characteristics of the sample grouped by sarcopenia status. Meds—medications; * Pearson’s chi-squared test. ^†^ Fisher’s exact test.

	No Sarcopenia (91/85%)	Sarcopenia (16/15%)	*p*-Value
Age Group (*n*/%)			0.143 ^†^
50–64 years	30 (90.9)	3 (9.1)	
65–74 years	34 (89.5)	4 (10.5)	
75+ years	27 (75.0)	9 (25.0)	
Women (*n*/%)	52 (57.1)	9 (56.3)	0.947 *
Hypertension (*n*/%)	42 (46.2)	9 (56.3)	0.456 *
Diabetes (*n*/%)	15 (16.5)	1 (6.3)	0.457 ^†^
Cardiovascular Meds (*n*/%)	44 (48.4)	10 (62.5)	0.297 *
Psychotropic Meds (*n*/%)	27 (29.7)	7 (43.8)	0.265 *

**Table 2 jcm-13-00018-t002:** Beta coefficients (β) and 95% confidence intervals (CIs) for the effect of sarcopenia status on the change in mean arterial pressure (MAP; mmHg), cardiac output (CO; L/min), total peripheral resistance (TPR; mmHg·s/mL), stroke volume (SV; mL), and heart rate (HR; beats per minute) from baseline at time intervals 0–10 s, 10–20 s, 20–30 s, 30–40 s, and 40–180 s after standing. Models were adjusted for age, sex, diabetes, hypertension, cardiovascular, and psychotropic medications. *** *p* < 0.001.

	0–10 s β (95% CI)	10–20 s β (95% CI)	20–30 s β (95% CI)	30–40 s β (95% CI)	40–180 s β (95% CI)
MAP	0.23 (−0.13, 0.59)	−0.67 (−1.03, −0.31) ***	−0.05 (−0.41, 0.31)	−0.10 (−0.45, 0.31)	0.03 (−0.02, 0.08)
CO	−0.05 (−0.08, −0.03) ***	0.06 (0.03, 0.09) ***	0.01 (−0.02, 0.04)	0.01 (−0.02, 0.04)	<−0.01 (<−0.01, <0.01)
TPR	<0.01 (−0.01, 0.01)	−0.02 (−0.03, −0.01) ***	0.01 (<−0.01, 0.02)	−0.01 (−0.02, <0.01)	<0.01 (<−0.01, <0.01)
SV	−0.37 (−0.72, −0.02)	0.54 (0.19, 0.88) ***	0.18 (−0.17, 0.52)	0.09 (−0.25, 0.43)	−0.02 (−0.06, 0.02)
HR	−0.16 (−0.40, 0.08)	−0.15 (−0.39, 0.09)	0.03 (−0.21, 0.26)	0.02 (−0.22, 0.25)	−0.01 (−0.04, 0.01)

## Data Availability

Contingent on compliance with data protection legislation and ethical approval, data may be available from the corresponding author upon reasonable request.

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
