# Peer review of "Haemodynamic Parameters Underlying the Relationship between Sarcopenia and Blood Pressure Recovery on Standing"

_jcm, 2023, doi:10.3390/jcm13010018_

Round 1

Reviewer 1 Report

Comments and Suggestions for Authors

This is an interesting study in which the authors examined the association between sarcopenia and orthostatic response in subjects aged 50 years and more.

There are some issues that should be dealt with in order to improve the manuscript:

1.       Abstract: Please avoid too many abbreviations, especially in the conclusions section.

2.       Participants were recruited from the Falls and Syncope Unit at the Mercer’s Institute for Successful Ageing in St James’s Hospital, Dublin, Ireland. Were all participants patients at the clinic, i.e. they had been referred due to falls and/or syncope? If that is the case, the study is limited by that all subjects were patients with falls/syncope and not from the general population. This is important and should be clearly stated in the methods, conclusions and abstract.

3.       Depending on the answer to question 2: Are there data for a completely healthy response to orthostasis that could be compared to the current findings?

4.       The status of medications at the time of the orthostatic test is a limitation that has already been stated by the authors. However, do the authors know how many participants that had medications with potential influence of orthostatic response, and if these were different between the patients with sarcopenia and no sarcopenia?

Author Response

This is an interesting study in which the authors examined the association between sarcopenia and orthostatic response in subjects aged 50 years and more.

We thank the reviewer for taking to time to consider our paper for their comments and suggestions which have strengthened the paper.

There are some issues that should be dealt with in order to improve the manuscript:

  1. Abstract: Please avoid too many abbreviations, especially in the conclusions section.

Many thanks for your comment, this has been addressed.

  1. Participants were recruited from the Falls and Syncope Unit at the Mercer’s Institute for Successful Ageing in St James’s Hospital, Dublin, Ireland. Were all participants patients at the clinic, i.e. they had been referred due to falls and/or syncope? If that is the case, the study is limited by that all subjects were patients with falls/syncope and not from the general population. This is important and should be clearly stated in the methods, conclusions and abstract.

Thank you for this very important point that has been clarified:

 107 participants were recruited from patients attending a falls and syncope clinic…

Typically, participants had been referred to the unit by their General Practitioner due to a recent history of falls, syncope, pre-syncope, light-headedness, or dizziness.

Finally, as participants were recruited from patients referred to a falls and syncope unit, they may not be representative of the general population, thus limiting the generalizability of our results.

  1. Depending on the answer to question 2: Are there data for a completely healthy response to orthostasis that could be compared to the current findings?

To the best of our knowledge and on detailed searching of the literature, we are not aware of any large population studies that have examined the haemodynamic response to standing in terms of the parameters included in this study. A number of smaller studies of healthy adults are referenced in the discussion including [6] Sprangers et al 1991, [30] Tanaka et al, [31] Wang et al and [32] Wieling et al.

  1. The status of medications at the time of the orthostatic test is a limitation that has already been stated by the authors. However, do the authors know how many participants that had medications with potential influence of orthostatic response, and if these were different between the patients with sarcopenia and no sarcopenia?

This is an important issue and as such we have included the number of participants on cardiovascular medications and psychotropic medications in Table 1. 48% of those without sarcopenia were taking cardiovascular medications compared with 62% of those with sarcopenia. In terms of psychotropic medications 29% of those without sarcopenia vs 43% of with sarcopenia. Because of the importance of these medications on the BP response to standing and the differences between the groups, we controlled for both as covariates in the mixed-effects models.

Reviewer 2 Report

Comments and Suggestions for Authors

We thank the Authors for their work, which provides an interesting insight into the mechanisms underlying altered responses to blood pressure adaptation in patients affected by sarcopenia. The paper is well written and adequately circumstantiated. I have only a single question to ask:

Ln 70: please explain the rationale for this specific age range. Dynapenia and sarcopenia are especially common and remarkably crucial in geriatric subjects (age 65+, per the most common definition), why include "younger" patients? What is the numerosity of the 50-64 y.o. subgroup?

Also, ln 38: typo, "[...]therefore to a reduction in intrathoracic[...]"

Author Response

We thank the Authors for their work, which provides an interesting insight into the mechanisms underlying altered responses to blood pressure adaptation in patients affected by sarcopenia. The paper is well written and adequately circumstantiated. I have only a single question to ask:

Many thanks for taking the time to consider our paper and for your constructive comments.

Ln 70: please explain the rationale for this specific age range. Dynapenia and sarcopenia are especially common and remarkably crucial in geriatric subjects (age 65+, per the most common definition), why include "younger" patients? What is the numerosity of the 50-64 y.o. subgroup?

Many thanks for this important comment. We agree that sarcopenia is increasingly common as age increases. In this study we aimed to study the evolution and effects of sarcopenia across a number of age groups. To make this clearer we have amended table 1 to include age groups 50-64, 65-74 and 75+. From this it can be seen there was a reasonable spread across the age groups n = 33, 38 and 36 respectively.

Also, ln 38: typo, "[...]therefore to a reduction in intrathoracic[...]"

This has been corrected.